electromagnetism/electrical engineering

transient analysis, power loss density, electromagnetic waves, Debye media

**Author for correspondence:**
Jiaqi Zhong
e-mail: plusingzhong@163.com

# Transient analysis of power loss density with time-harmonic electromagnetic waves in Debye media

Jiaqi Zhong[1], Shan Liang[2], Yong Chen[1] and Jiajia Tan[3]

[1]College of Automation, Chongqing University of Posts and Telecommunications, Chongqing 400065, People's Republic of China
[2]College of Automation, Chongqing University, Chongqing 400044, People's Republic of China
[3]College of Traffic and Transportation, Chongqing Jiaotong University, Chongqing 400074, People's Republic of China

JZ, 0000-0003-3471-0580

Due to the complex permittivity, it is difficult to directly clarify the transient mechanism between electromagnetic waves and Debye media. To overcome the above problem, the temporal relationship between the electromagnetic waves and permittivity is explicitly derived by applying the Fourier inversion and introducing the remnant displacement. With the help of the Poynting theorem and energy conservation equation, the transient power loss density is derived to describe the transient dissipation of electromagnetic field and the mechanism on phase displacement has been explicitly revealed. Besides, the unique solution can be obtained by applying the time-domain analysis method rather than involving the frequency-domain characteristics. The effectiveness of transient analysis is demonstrated by giving a comparison simulation on one-dimensional example.

## 1. Introduction

Over the past few decades, the applications of electromagnetic energy [1,2] have received much attention due to the high-efficiency, energy-saving and pollution-free characteristics. As a novel energy carrier, the application value has shown great prospects [3,4] in domestic and industrial fields. With the excellent penetration effect in Debye media, the electromagnetic waves can lead to the high-frequency oscillation of molecule [5] or molecular clusters [6]. Nevertheless, it is still difficult to clarify the interactions between the electromagnetic field and Debye media, especially in chemical reactions process.

From the view of classical electromagnetic theory, the local dissipation energy [7,8] can be described by Poynting's theorem

and conservation of energy. However, it is still difficult to provide the analytical expression of power loss density in time domain due to the complex permittivity. To overcome the above problem, Maxwell's curl equation [9,10] is usually applied to describe the propagation of electromagnetic field in frequency domain. Although the frequency analysis method [11] provides an effective way to estimate the dissipation of electromagnetic energy in the industrial field, the cumbersome method cannot describe the dynamic characteristics in transient state. Especially in the microwave chemistry field, the permittivity is always temperature-dependent and frequency-dependent [12] until the reactions reach equilibrium. Obviously, it is possibly inappropriate for the classical frequency or time-frequency domain model to describe the transient mechanism between the electromagnetic field and Debye media. Therefore, the time-domain analysis of power loss density contributes to clarify the coupling relationship between electromagnetic waves and Debye media.

It is worth pointing out that the relationship [13] between the electric field and displacement current is usually applied to describe the transient power loss density for Debye media. Although many researchers, such as, Converse, Gandhi and Torres *et al.* [14–16], propose the different expressions to describe the same process, the unique solution is obtained by applying the frequency-dependent finite-difference time-domain ((FD)²TD) method. The computational burden will be increased due to the discrete convolution. Kobidze *et al.* [17] analyses the transient scattering from inhomogeneous dispersive bodies based on a fast time-domain integral equation. Uysal *et al.* [18] develop a time-domain surface integral equation solver for analysing electromagnetic field interactions on plasmonic nanostructures. Sayed *et al.* [19] propose an explicit marching-on-in-time scheme to describe the nonlinear and dispersive scatterers. With the electrodynamic approach [13,20] and equivalent circuit approach [21], a power loss density [22] is derived to overcome the constraint of frequency domain, but the coupling mechanism between the electromagnetic field and Debye media is still not be explicitly revealed.

For time-harmonic electromagnetic waves, the law of ohmic losses can describe the time-averaged ohmic power losses per unit volume, but it still cannot provide the transient process of electromagnetic propagation in Debye media. To the best of our knowledge, these are few results which can directly provide the time-domain solution of power loss density based on Maxwell's equation. Motivated by above problem, applying Fourier inversion and introducing the remnant displacement can explicitly clarify the temporal mechanism of electromagnetic dissipation in the Debye media. With the help of energy conservation equation, the simple transient power loss density can be explicitly proposed. Besides, the finite-difference time-domain (FDTD) method is applied to solve the coupled Maxwell's equation and ordinary differential equation (ODE). Finally, a comparison simulation on one-dimensional example demonstrates the proposed methodology is effective.

## 2. Transient power loss density in Debye media

Considering the source-free, linear, isotropic and non-magnetic Debye media, the general formulation of time-dependent Maxwell's equations can be given as

$$\nabla \times \mathbf{H}(t) = \frac{\partial \mathbf{D}(t)}{\partial t}, \tag{2.1}$$

$$\mathbf{D}(\omega) = \varepsilon_0 \left( \varepsilon_\infty + \frac{\varepsilon_s - \varepsilon_\infty}{1 + j\omega\tau} \right) \mathbf{E}(\omega) \tag{2.2}$$

and

$$\nabla \times \mathbf{E}(t) = -\mu_0 \frac{\partial \mathbf{H}(t)}{\partial t}, \tag{2.3}$$

where the quantities $\mathbf{H}(t)$ and $\mathbf{E}(t)$ denote the magnetic and electric field intensity, respectively; $\mathbf{D}$ is the electric flux density; $\varepsilon_0$, $\mu_0$, $\varepsilon_s$ and $\varepsilon_\infty$ are the vacuum permittivity, vacuum permeability, static relative permittivity and the relative permittivity at the infinite frequency, respectively; $\omega$ and $\tau$ denote the angular frequency and the relaxation time.

It follows that Maxwell's wave equations (2.1) and (2.3) can describe the transient characteristics in the spatio-temporal domain. However, the relationship (2.2) between electric field intensity and electric flux density is always represented in frequency domain instead of time domain, due to the frequency-dependent characteristics. Obviously, it is difficult to directly analyse the transient characteristics in time-frequency domain. In order to derive the transient power loss density, the electric flux density (2.2) can be expressed as

$$(1 + j\omega\tau)\mathbf{D}(\omega) = \varepsilon_0(j\omega\varepsilon_\infty\tau + \varepsilon_s)\mathbf{E}(\omega). \tag{2.4}$$

**Remark 2.1.** The traditional method usually applies the relative permittivity $\varepsilon'$ and relative dielectric loss $\varepsilon''$ to simplify (2.2), which can be expressed as $\mathbf{D}(\omega) = \varepsilon_0\varepsilon'\mathbf{E}(\omega) - j\varepsilon_0\varepsilon''\mathbf{E}(\omega)$. By applying the inverse Fourier transform, (2.1) can be transformed as $\nabla \times \mathbf{H}(t) = \varepsilon_0\varepsilon'(\partial\mathbf{E}(t)/\partial t) + \omega\varepsilon_0\varepsilon''\mathbf{E}(t)$. Strictly speaking, $\varepsilon'$ and $\varepsilon''$ are dependent frequency. For the dispersive media, applying above method may lead the loss of dynamical characteristics. Therefore, it is important for the transient analysis of electromagnetic field to obtain the temporal relationship between $\mathbf{D}(\omega)$ and $\mathbf{E}(\omega)$.

Substituting $\mathbf{D}(\omega) = \varepsilon_0\varepsilon_\infty\mathbf{E}(\omega) + \mathbf{D}_r(\omega)$ into (2.4), we have

$$j\omega\tau\mathbf{D}_r(\omega) + \mathbf{D}_r(\omega) = \varepsilon_0(\varepsilon_s - \varepsilon_\infty)\mathbf{E}(\omega), \tag{2.5}$$

where $\mathbf{D}_r(\omega)$ denotes the remnant displacement. With the help of inverse Fourier transform, (2.5) can be transformed as

$$\tau\frac{\partial\mathbf{D}_r(t)}{\partial t} + \mathbf{D}_r(t) = \varepsilon_0(\varepsilon_s - \varepsilon_\infty)\mathbf{E}(t) \tag{2.6}$$

Obviously, (2.6) indicates that the auxiliary parameter $\mathbf{D}_r$ is hysteretic parameter which depends on $\varepsilon_\infty$, $\varepsilon_s$ and $\tau$. Based on Poynting's power balance theorem, combining (2.1), (2.3) and (2.6) yields

$$
\begin{aligned}
-\nabla \times (\mathbf{E} \times \mathbf{H}) &= \mathbf{E}(t)\frac{\partial\mathbf{D}(t)}{\partial t} + \mu_0\mathbf{H}(t)\frac{\partial\mathbf{H}(t)}{\partial t} \\
&= \varepsilon_0\varepsilon_\infty\mathbf{E}(t)\frac{\partial\mathbf{E}(t)}{\partial t} + \mathbf{E}(t)\frac{\partial\mathbf{D}_r(t)}{\partial t} + \mu_0\mathbf{H}(t)\frac{\partial\mathbf{H}(t)}{\partial t} \\
&= \varepsilon_0\varepsilon_\infty\mathbf{E}(t)\frac{\partial\mathbf{E}(t)}{\partial t} + \mu_0\mathbf{H}(t)\frac{\partial\mathbf{H}(t)}{\partial t} \\
&\quad + \left(\frac{\tau}{\varepsilon_0(\varepsilon_s - \varepsilon_\infty)}\frac{\partial\mathbf{D}_r(t)}{\partial t} + \frac{1}{\varepsilon_0(\varepsilon_s - \varepsilon_\infty)}\mathbf{D}_r(t)\right)\frac{\partial\mathbf{D}_r(t)}{\partial t}.
\end{aligned} \tag{2.7}
$$

In order to obtain the explicit power loss density, the energy conservation equation [21] can be expressed as

$$-\nabla \times (\mathbf{E} \times \mathbf{H}) = \frac{\partial W_e}{\partial t} + \frac{\partial W_b}{\partial t} + P_{\text{loss}}, \tag{2.8}$$

where $W_e$ and $W_b$ denote the electric energy density and magnetic energy density, respectively. $P_{\text{loss}}$ is the power loss density. By comparing (2.7) and (2.8), we have

$$W_e = \varepsilon_0\varepsilon_\infty\frac{|\mathbf{E}(t)|^2}{2} + \frac{1}{\varepsilon_0(\varepsilon_s - \varepsilon_\infty)}\frac{|\mathbf{D}_r(t)|^2}{2}, \tag{2.9}$$

$$W_b = \mu_0\frac{|\mathbf{H}(t)|^2}{2} \tag{2.10}$$

and

$$P_{\text{loss}} = \frac{\tau}{\varepsilon_0(\varepsilon_s - \varepsilon_\infty)}\left|\frac{\partial\mathbf{D}_r(t)}{\partial t}\right|^2. \tag{2.11}$$

**Remark 2.2.** For the single-pole Debye media [23], the power loss density depends on the remnant displacement $\mathbf{D}_r(t)$ instead of the $\mathbf{E}(t)$. However, the solution of (2.6) is $\varepsilon_0(\varepsilon_s - \varepsilon_\infty)\mathbf{E}(t)/\tau \cdot e^{-t/\tau}$, which shows that $\mathbf{D}_r(t)$ lags behind $\mathbf{E}(t)$. On the contrary, the derived power loss density $P_{\text{loss}}$ has different characteristics. Traditionally, the power loss density can be expressed as

$$P'_{\text{loss}} = \omega\varepsilon_0\varepsilon''\mathbf{E}^2(t). \tag{2.12}$$

It follows from (2.12) that the derived power loss density (2.11) is based on the derivation of $\mathbf{D}(\omega)$. In order to explicitly demonstrate the characteristics, defining the auxiliary polarization current as $\mathbf{J}_p = \partial\mathbf{D}_r(t)/\partial t$, the derivative of (2.6) can be transformed as

$$\tau\frac{\partial\mathbf{J}_p(t)}{\partial t} + \mathbf{J}_p(t) = \varepsilon_0(\varepsilon_s - \varepsilon_\infty)\frac{\partial\mathbf{E}(t)}{\partial t}. \tag{2.13}$$

The solution of (2.13) is $\mathbf{J}_p = \varepsilon_0(\varepsilon_s - \varepsilon_\infty)\mathbf{E}(t)/\tau \cdot (1 - e^{-t/\tau})$, which means the derived power loss density will lead the traditional one.

**Remark 2.3.** From the view of frequency domain, the dissipation power depends on the imaginary part of (2.2). The definition $D_r(t)$ involves a part of relative permittivity $\varepsilon'$, which may change the propagation of electromagnetic field. Therefore, electric energy density in (2.8) relies on the $E(t)$ and $D_r(t)$. Different with the Lorentz and Drude media, the Debye media is non-magnetic, which means that magnetic energy density (2.10) only depends on $H(t)$.

**Remark 2.4.** It is worth pointing out that the classical power loss density is based on the analysis of frequency domain, which will inevitably ignore some transient dynamical characteristics. However, the dissipation of power is the inherent characteristics for the Debye media, which means that the total energy of loss power is the same whether time-domain or frequency-domain analysis. Therefore, the above methodology has considered the whole dynamical characteristics which will contribute to more accurately reveal the law of electromagnetic propagation in the Debye media.

# 3. Numerical analysis

## 3.1. Implement with FDTD

Based on the aforementioned analysis, the coupled Maxwell's equation and ODE (2.1), (2.3) and (2.6) can describe the transient process of electromagnetic propagation in Debye media. Substituting (2.6) into (2.1) yields

$$
\nabla \times H = \varepsilon_0 \varepsilon_\infty \frac{\partial E(t)}{\partial t} + \frac{\partial D_r(t)}{\partial t}
$$
$$
= \varepsilon_0 \varepsilon_\infty \frac{\partial E(t)}{\partial t} + J_p(t). \tag{3.1}
$$

In order to overcome the obstruction of two temporal differential operators, the definition $J_p$ will transform Maxwell's equation into a feasible expression. By combining (2.3), (2.13) and (3.1), the electromagnetic intensity can be solved by applying the FDTD method [24]. For explicitly illustrating the proposed methodology, we especially give a one-dimensional example, whose expression can be simplified as

$$
\varepsilon_0 \varepsilon_\infty \frac{\partial E_x(t)}{\partial t} = -\frac{\partial H_y(t)}{\partial z} - J_{px}(t) \tag{3.2}
$$

and

$$
\frac{\partial H_y(t)}{\partial t} = -\frac{1}{\mu_0} \frac{\partial E_x(t)}{\partial z}. \tag{3.3}
$$

From (3.2) and (3.3), we denote that the plane wave propagates in the $z$-direction, whose electric field orients with the $x$-direction and magnetic field orients with the $y$-direction. By applying the forward difference approximations for both the temporal and spatial differential operators, (3.2) and (3.3) can be transformed as

$$
\varepsilon_0 \varepsilon_\infty \frac{E_x^{n+1}(k) - E_x^n(k)}{\Delta t} = -\frac{H_y^n(k) - H_y^n(k-1)}{\Delta z} - J_{px}^n(k), \tag{3.4}
$$

$$
\frac{H_y^{n+1}(k) - H_y^n(k)}{\Delta t} = -\frac{1}{\mu_0} \frac{E_x^n(k+1) - E_x^n(k)}{\Delta z} \tag{3.5}
$$

and
$$
\tau \frac{J_{px}^{n+1}(k) - J_{px}^n(k)}{\Delta t} + J_{px}^n(k) = \varepsilon_0 (\varepsilon_s - \varepsilon_\infty) \frac{E_x^{n+1}(k) - E_x^n(k)}{\Delta t}, \tag{3.6}
$$

where $n$ and $k$ indicate a time step $t = \Delta t \cdot n$ and a spatial distance $z = \Delta z \cdot k$. Based on the method of numerical recursion, we can transform (3.4) and (3.6) into the following computable expression:

$$
E_x^{n+1}(k) = E_x^n(k) - \frac{\Delta t}{\varepsilon_0 \varepsilon_\infty \Delta z} (H_y^n(k) - H_y^n(k-1)) - \frac{\Delta t}{\varepsilon_0 \varepsilon_\infty} J_{px}^n(k), \tag{3.7}
$$

$$
H_y^{n+1}(k) = H_y^n(k) - \frac{\Delta t}{\mu_0 \Delta z} (E_x^n(k+1) - E_x^n(k)) \tag{3.8}
$$

and
$$
J_{px}^{n+1}(k) = \left(1 - \frac{\Delta t}{\tau}\right) J_{px}^n(k) + \frac{\varepsilon_0 (\varepsilon_s - \varepsilon_\infty)}{\tau} (E_x^{n+1}(k) - E_x^n(k)). \tag{3.9}
$$

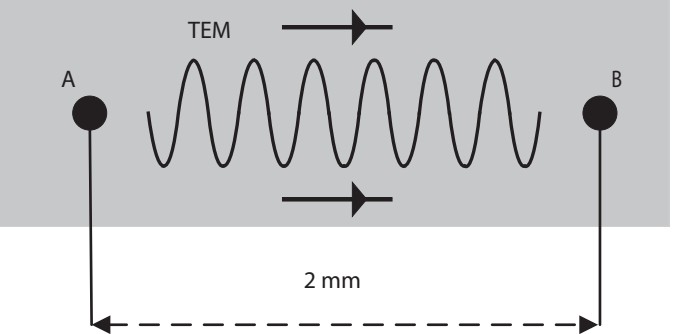

**Figure 1.** Schematic diagram of TEM plane wave in Debye media.

**Remark 3.1.** In the traditional method, applying the central difference approximations discretizes the temporal and spatial differential operators in Maxwell's equation (i.e. (2.1) and (2.3)) to obtain the spatio-temporal characteristics. In the transient analysis, the solution of coupled Maxwell's equation and ODE can be obtained by applying the forward or backward difference method. Therefore, the proposed method will increase the burden of computation for the transient analysis.

Although the coupled Maxwell's equation and ODE model (2.13), (3.2) and (3.3) has been transformed as the ODEs model (3.7)–(3.9), the simulation may lead to the incorrect results in the case of inappropriate time steps. With the help of Courant–Friedrichs–Lewy condition [25], the discrete time steps of equal duration must be less than the time for the wave to travel to adjacent grid points. For the above one-dimensional case, the speed of electromagnetic wave in Debye media cannot go faster than the speed of light in free space. We can define the following form:

$$\frac{\Delta t}{\Delta x} \leq \frac{\sqrt{\mu_0 \varepsilon_0}}{C_N},$$

(3.10)

where the Courant number $C_N$ is typically not less than 1.

## 3.2. Simulation

To further analyse the transient power loss density (2.11), we first assume the Debye media exposed in traverse electromagnetic (TEM) plane wave, which is shown in figure 1 and the incident electromagnetic pulse in the point A can be defined as

$$E(t) = \sin(2\pi f t),$$

(3.11)

where $f$ is 20 GHz. The left and right boundary can be defined as the perfectly matched layer, which indicates that the residual energy will be absorbed totally. For the propagation medium, we choose a well-known Debye medium, i.e. deionized water, whose static permittivity [15] can be expressed as

$$\varepsilon_s = \frac{3\varepsilon_\infty T + A(\varepsilon_\infty + 2)^2 + \sqrt{[3\varepsilon_\infty T + A(\varepsilon_\infty + 2)^2]^2 + 72\varepsilon_\infty^2 T^2}}{12T}$$

(3.12)

where $A = A_0 e^{-U/kT}$ and $T$ is the local temperature, which can be assumed as the 20°C.

The relaxation time can be rewritten as

$$\tau = \tau_0 e^{W_a/KT},$$

(3.13)

where $\tau_0$ is the constant of the medium, $W_a$ denotes the activation energy and $K$ is generally known as Boltzmann's constant. All the dielectric parameters of deionized water [15] are given in table 1.

As shown in figure 1, we will analyse the power loss density at the point B, which is located in the right hand of 2 mm. Before the simulation, we first denote the mesh size is 0.1 mm, the time step is 0.01 ps and the total time is 0.35 ns. With the help of FDTD method, the comparison results on the different transient power loss density (i.e. (2.11) and (2.12)) are presented in figure 2.

When the electromagnetic field reaches the point B, the arrangement of polar molecule will be broken. The amplitudes and phases of electromagnetic wave will be changed by a large margin until the coupling process reaches equilibrium. As time goes on, the transient response of electromagnetic field will be

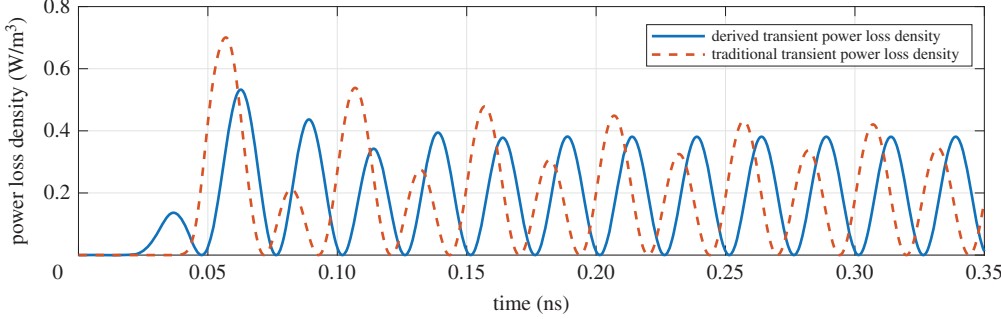

**Figure 2.** Comparison between the derived and traditional transient power loss density.

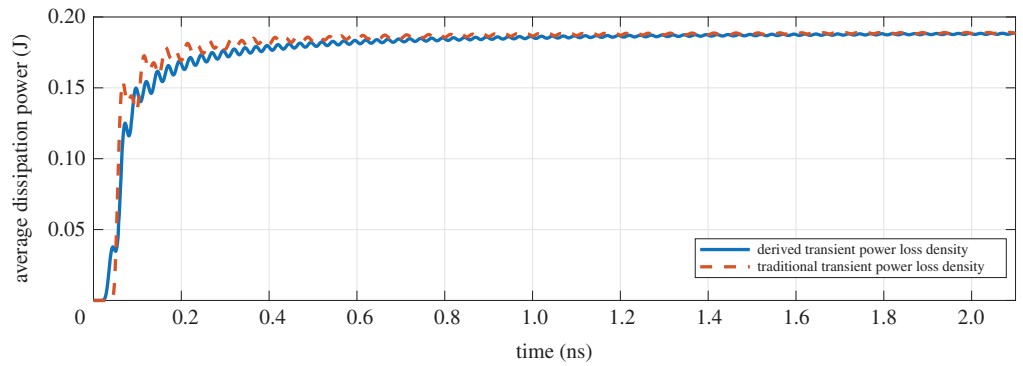

**Figure 3.** Comparison on the average dissipation power based on the different transient power densities.

**Table 1.** Dielectric parameters of deionized water.

| $\varepsilon_\infty$ | $A_0$ | $U$ | $\tau_0$ | $W_a$ |
|---|---|---|---|---|
| 5.5 | 1186.78 | $-2.88 \times 10^{-21}$ J | $6.27 \times 10^{-15}$ s | $2.96 \times 10^{-20}$ J |

transformed as the steady response, which has the fixed amplitude and phase. In terms of (2.11) and (2.12), the two transient power loss densities can reveal the above unsteady phenomenon. But (2.11) can reach the steady state at the relatively short time as shown in figure 2, because any dynamical characteristics are not ignored in time-domain analysis. Moreover, the leading amplitude problem in remark 2.3 has been demonstrated in this simulation. The leading time is almost equal to the relaxation time $\tau$, i.e. 0.00941 ns.

In order to further demonstrate the validity, the dissipation power in the point B can be obtained by applying the integral definition for the derived and traditional power loss density, respectively. As shown in figure 3, the non-equilibrium state has the different average dissipation powers due to irregular oscillation, which facilitates the absorption of more electromagnetic energy and accelerates the rearrangement of dipole. Once the electromagnetic field in the medium approaches the equilibrium state, the dissipation powers derived by the different methods will reach an excellent agreement, which also demonstrates the explanation of remark 2.4. Therefore, the effectiveness of transient power loss density has been successfully validated.

# 4. Conclusion

In this paper, we explicitly reveal the coupling relationship between the electromagnetic waves and Debye media in time domain. Based on the time-harmonic characteristics and Fourier inversion, the transient mechanism of electromagnetic field and electric flux density is revealed in time-domain analysis. With Poynting's theorem and energy conservation equation, a simple expression is proposed to describe the

power loss density. In addition, the comparison results on the transient power loss density and average dissipation power indicate that the proposed transient expression is effective.

Data accessibility. All code are available from the Dryad Digital Repository: https://datadryad.org/stash/dataset/doi:10.5061/dryad.547d7wm79 [26].

Authors' contributions. J.Z. is the first and corresponding author, who mainly engages in the theoretical derivation and numerical simulation. S.L. mainly provides an idea about the transient analysis of electromagnetic waves. Y.C. mainly compares between the traditional method and proposed method. J.T. mainly gives the review of references and polishes the language of manuscript.

Competing interests. We declare we have no competing interests.

Funding. This work was supported by the National Natural Science Foundation of China (62003066, 61771077), the Science and Technology Research Program of Chongqing Municipal Education Commission (KJQN201900614), the Natural Science Foundation of Chongqing (cstc2021jcyj-msxmX0331) and the China Scholarship Council (202008500013).

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
