## [Peer Review File · Royal Society Open Science]

Review History

RSOS-210023.R0 (Original submission)

Review form: Reviewer 1

Is the manuscript scientifically sound in its present form?

Yes

Are the interpretations and conclusions justified by the results?

Yes

Is the language acceptable?

No

Do you have any ethical concerns with this paper?

No

Have you any concerns about statistical analyses in this paper?

No

Recommendation?

Accept with minor revision (please list in comments)

Comments to the Author(s)

- 1) English should be improved.
- 2) Please show an example to compare the proposed method to the frequency analysis more details.
- 3) What's the condition for obtaining a unique solution ?
- 4) Please explain the affect from the central difference approximations for (3.1), (3.2) in theory and simulation.

Review form: Reviewer 2**Is the manuscript scientifically sound in its present form?**

No

Are the interpretations and conclusions justified by the results?

Yes

Is the language acceptable?

Yes

Do you have any ethical concerns with this paper?

No

Have you any concerns about statistical analyses in this paper?

No

Recommendation?

Reject

Comments to the Author(s)

This paper presents a novel formulation and solution procedure for the time domain analysis of electromagnetic wave interaction with Debye media. It stated that there is a problem on the determination of the dissipated energy for in the Debye media in frequency domain. It is claimed that due to the time dependent permittivity of the Debye media it is not possible to use frequency domain methods directly.

Unfortunately I recommend the rejection of the paper because the paper does not sufficiently advances the scientific knowledge.

The problem presented in this paper is very well known, where the energy relations including instantaneous Poynting vector and its time harmonic counterpart can be found in [R1], and the analysis of the Debye media and other dispersive media using FDTD can be found in [R2], and there are other time domain methods that is developed for the transient analysis of dispersive media e.g. in [R3]-[R5]. Similar to the authors, anyone can obtain the transient solution of electric or magnetic field using the methods given in [R2]-[R5] and use the energy relations in [R1] to determine dissipated energy.

Moreover, the formulation presented in the paper is only valid for the time harmonic excitation. The equations that has both frequency and time domain expressions are not valid for general cases. This can be seen in the given example for a single sinusoidal excitation. As a different point of view, authors showed that if the excitation has a sinusoidal behavior, frequency domain, in other words time harmonic, analysis provides the solution for steady state case. There is no need for a time domain simulation.

Other comments to the authors for possible future publications:

- 1-Using terms "modified Maxwell's equations" or "modifying Maxwell's equations" can be misunderstood. The authors are not modifying the Maxwell's equations, instead they are inserting the constitutive relations for the Debye media, which can be considered as specifying the polarization vector in Maxwell's equations, and they are mixing the time domain, frequency domain, and time harmonic cases.
- 2-For a dispersive medium, dielectric constant is not constant, therefore it would be better to call it permittivity.
- 3-It should be mentioned that the equation (2.8) is given for source free region.
- 4-Equation (2.9) is given for instantaneous time [R1].
- 5-In figures 2 and 3, the region authors called "the unstable electromagnetic wave" is actually called transient region and time domain analysis is required (mostly) to determine this transient response.
- 6-Please include an example that has an actual pulse excitation as in [R3].
- 7-The authors use $\exp(j\omega t)$ time dependence in Section 2 and in equations (3.12) and (3.13) they switched to $\exp(-i\omega t)$ time dependence without noticing.
- 8-It is always better to present an example that compares the proposed method with a well-known method e.g. FDTD method as in [R2].

References:

- [R1] Balanis, Constantine A. Advanced Engineering Electromagnetics. New York: Wiley, 1989.
- [R2] Taflove, Allan and Hagness, Susan C.. Computational Electrodynamics: The Finite-Differences Time-Domain Method. 2 Boston, London: Artech House, Inc., 2000.
- [R3] G. Kobidze, Jun Gao, B. Shanker and E. Michielssen, "A fast time domain integral equation based scheme for analyzing scattering from dispersive objects," in IEEE Transactions on Antennas and Propagation, vol. 53, no. 3, pp. 1215-1226, March 2005, doi: 10.1109/TAP.2004.841295.
- [R4] Ismail E. Uysal, et al., "Quantum-corrected transient analysis of plasmonic nanostructures," Opt. Express 25, 5891-5908 (2017)
- [R5] S. B. Sayed, H. A. Ulku and H. Bagci, "An explicit MOT scheme for solving the TD-EFVIE on nonlinear and dispersive scatterers," 2017 IEEE International Symposium on Antennas and Propagation & USNC/URSI National Radio Science Meeting, 2017, pp. 1135-1136, doi: 10.1109/APUSNCURSINRSM.2017.8072610.

Decision letter (RSOS-210023.R0)

Dear Dr Zhong

The Editors assigned to your paper RSOS-210023 "Transient Analysis of Power Loss Density with Time-Harmonic Electromagnetic Waves in Debye Media" have now received comments from reviewers and would like you to revise the paper in accordance with the reviewer comments and

any comments from the Editors. Please note this decision does not guarantee eventual acceptance.

Please submit your revised manuscript and required files (see below) no later than 21 days from today's (ie 18-May-2021) date. Note: the ScholarOne system will 'lock' if submission of the revision is attempted 21 or more days after the deadline. If you do not think you will be able to meet this deadline please contact the editorial office immediately.

on behalf of Dr Peter Munro (Associate Editor) and Miles Padgett (Subject Editor)
openscience@royalsociety.org

Associate Editor Comments to Author (Dr Peter Munro):

Comments to the Author:

Please take note of reviewer comments which relate to a lack of technical soundness.

Reviewer comments to Author:

Reviewer: 1

Comments to the Author(s)

1) English should be improved.

2) Please show an example to compare the proposed method to the frequency analysis more details.

3) What's the condition for obtaining a unique solution ?

4) Please explain the affect from the central difference approximations for (3.1), (3.2) in theory and simulation.

Reviewer: 2

Comments to the Author(s)

This paper presents a novel formulation and solution procedure for the time domain analysis of electromagnetic wave interaction with Debye media. It stated that there is a problem on the determination of the dissipated energy for in the Debye media in frequency domain. It is claimed that due to the time dependent permittivity of the Debye media it is not possible to use frequency domain methods directly.

Unfortunately I recommend the rejection of the paper because the paper does not sufficiently advances the scientific knowledge.

The problem presented in this paper is very well known, where the energy relations including instantaneous Poynting vector and its time harmonic counterpart can be found in [R1], and the analysis of the Debye media and other dispersive media using FDTD can be found in [R2], and there are other time domain methods that is developed for the transient analysis of dispersive media e.g. in [R3]-[R5]. Similar to the authors, anyone can obtain the transient solution of electric or magnetic field using the methods given in [R2]-[R5] and use the energy relations in [R1] to determine dissipated energy.

Moreover, the formulation presented in the paper is only valid for the time harmonic excitation. The equations that has both frequency and time domain expressions are not valid for general cases. This can be seen in the given example for a single sinusoidal excitation. As a different point of view, authors showed that if the excitation has a sinusoidal behavior, frequency domain, in other words time harmonic, analysis provides the solution for steady state case. There is no need for a time domain simulation.

Other comments to the authors for possible future publications:

- 1-Using terms "modified Maxwell's equations" or "modifying Maxwell's equations" can be misunderstood. The authors are not modifying the Maxwell's equations, instead they are inserting the constitutive relations for the Debye media, which can be considered as specifying the polarization vector in Maxwell's equations, and they are mixing the time domain, frequency domain, and time harmonic cases.
- 2-For a dispersive medium, dielectric constant is not constant, therefore it would be better to call it permittivity.
- 3-It should be mentioned that the equation (2.8) is given for source free region.
- 4-Equation (2.9) is given for instantaneous time [R1].
- 5-In figures 2 and 3, the region authors called "the unstable electromagnetic wave" is actually called transient region and time domain analysis is required (mostly) to determine this transient response.
- 6-Please include an example that has an actual pulse excitation as in [R3].
- 7-The authors use $\exp(j\omega t)$ time dependence in Section 2 and in equations (3.12) and (3.13) they switched to $\exp(-i\omega t)$ time dependence without noticing.
- 8-It is always better to present an example that compares the proposed method with a well-known method e.g. FDTD method as in [R2].

References:

- [R1] Balanis, Constantine A. Advanced Engineering Electromagnetics. New York: Wiley, 1989.
- [R2] Taflove, Allan and Hagness, Susan C.. Computational Electrodynamics: The Finite-Differences Time-Domain Method. 2 Boston, London: Artech House, Inc., 2000.
- [R3] G. Kobidze, Jun Gao, B. Shanker and E. Michielssen, "A fast time domain integral equation based scheme for analyzing scattering from dispersive objects," in IEEE Transactions on Antennas and Propagation, vol. 53, no. 3, pp. 1215-1226, March 2005, doi: 10.1109/TAP.2004.841295.

[R4] Ismail E. Uysal, et al., "Quantum-corrected transient analysis of plasmonic nanostructures," *Opt. Express* 25, 5891-5908 (2017)

[R5] S. B. Sayed, H. A. Ulku and H. Bagci, "An explicit MOT scheme for solving the TD-EFVIE on nonlinear and dispersive scatterers," 2017 IEEE International Symposium on Antennas and Propagation & USNC/URSI National Radio Science Meeting, 2017, pp. 1135-1136, doi: 10.1109/APUSNCURSINRSM.2017.8072610.

===PREPARING YOUR MANUSCRIPT===

===PREPARING YOUR REVISION IN SCHOLARONE===

Author's Response to Decision Letter for (RSOS-210023.R0)

See Appendix A.

RSOS-210023.R1 (Revision)

Review form: Reviewer 1

Is the manuscript scientifically sound in its present form?

Yes

Are the interpretations and conclusions justified by the results?

Yes

Is the language acceptable?

Yes

Do you have any ethical concerns with this paper?

No

Have you any concerns about statistical analyses in this paper?

No

Recommendation?

Accept as is

Comments to the Author(s)

The response letter is good. This manuscript can be acceptable.

Review form: Reviewer 3

Is the manuscript scientifically sound in its present form?

Yes

Are the interpretations and conclusions justified by the results?

No

Is the language acceptable?

Yes

Do you have any ethical concerns with this paper?

No

Have you any concerns about statistical analyses in this paper?

No

Recommendation?

Reject

Comments to the Author(s)

The authors claim two points: a) this article has derived the transient power loss density of Debye media under exposure of EM waves based on of the Poynting theorem and energy conservation equation and has given the mechanism on phase displacement; and b) the unique solution by

applying the time-domain analysis method rather than involving the frequency-domain characteristics.

In my viewpoint, it is an interesting topic, because the transient power loss density is crucial to understand how the energy transform into heat and build theoretical model of microwave heating. However, as far as I know, they are not new results. My detailed comments are

1. The main results in the manuscript are included in equations (2.5) -(2.11). These results can also be found in equations (2.4) -(2.11) in reference [22]. All the derivations are nearly the same. Besides, reference [22] has proved that this power loss density is caused by the resistance, which clearly shows the mechanism.
2. This time-domain analysis method is also as the same as the FDTD in the Taflove's textbook. How to couple the dispersive media, such as Debye, Lorentz, Drude media, to Maxwell's equations in FDTD method is well developed. So, it is not new.

Decision letter (RSOS-210023.R1)

Dear Dr Zhong,

It is a pleasure to accept your manuscript entitled "Transient Analysis of Power Loss Density with Time-Harmonic Electromagnetic Waves in Debye Media" in its current form for publication in Royal Society Open Science. The comments of the reviewer(s) who reviewed your manuscript are included at the foot of this letter.

Please see the Royal Society Publishing guidance on how you may share your accepted author manuscript at <https://royalsociety.org/journals/ethics-policies/media-embargo/>. After

publication, some additional ways to effectively promote your article can also be found here <https://royalsociety.org/blog/2020/07/promoting-your-latest-paper-and-tracking-your-results/>.

on behalf of Dr Peter Munro (Associate Editor) and Miles Padgett (Subject Editor)
openscience@royalsociety.org

Reviewer comments to Author:

Reviewer: 1

Comments to the Author(s)

The response letter is good. This manuscript can be acceptable.

Reviewer: 3

Comments to the Author(s)

The authors claim two points: a) this article has derived the transient power loss density of Debye media under exposure of EM waves based on of the Poynting theorem and energy conservation equation and has given the mechanism on phase displacement; and b) the unique solution by applying the time-domain analysis method rather than involving the frequency-domain characteristics.

In my viewpoint, it is an interesting topic, because the transient power loss density is crucial to understand how the energy transform into heat and build theoretical model of microwave heating. However, as far as I know, they are not new results. My detailed comments are

1. The main results in the manuscript are included in equations (2.5) -(2.11). These results can also be found in equations (2.4) -(2.11) in reference [22]. All the derivations are nearly the same. Besides, reference [22] has proved that this power loss density is caused by the resistance, which clearly shows the mechanism.
2. This time-domain analysis method is also as the same as the FDTD in the Taflove's textbook. How to couple the dispersive media, such as Debye, Lorentz, Drude media, to Maxwell's equations in FDTD method is well developed. So, it is not new.

Appendix A

Dear Dr. Peter Munro,

Thank you very much for your letter with regard to our manuscript (ID RSOS-210023) together with the comments from two reviewers. According to the comments, we have tried our best to check the paper and address all the comments from reviewers. And we are very sorry to submit the revised manuscript late. Because we have found the derived power loss density will lead the traditional one. It is an amazing result, which has spent much time to analyze and validate. The changes are highlighted in PDF file. Now, I am sending our revised manuscript to you.

Our incorporation of the reviewers' suggestions is as follows:

Point 1: English should be improved.

Response 1: We apologize for our carelessness. We have found a familiar-English colleague to correct the language in our manuscript. The readability of the manuscript has been developed.

Point 2: Please show an example to compare the proposed method to the frequency analysis more details.

Response 2: Thanks for your kind suggestion. By applying the FDTD method, we compare the derived and traditional power loss density. Therefore, the comparison results have been revised in the last version of the manuscript, as shown:

“Before the simulation, we first denote the mesh size is 0.1 mm, the time step is 0.01 ps and the total time is 0.35 ns. With the help of FDTD method, the comparison results on the different transient power loss density (i.e., (2.11) and (2.12)) can be presented in Figure 2.

Figure 2 Comparison between the derived and traditional transient power loss density

When the electromagnetic field reaches the point B, the arrangement of polar molecule will be broken. The amplitudes and phases of electromagnetic wave will be changed by a large margin until the coupling process reaches equilibrium. As time goes on, the transient response of electromagnetic field will be transformed as the steady response, which has the fixed amplitude and phase. In terms of (2.11) and (2.12), the two transient power loss densities can reveal the above unsteady phenomenon. But (2.11) can reach the steady state at the relatively short time as shown in Figure 2. Because any dynamical characteristics is not ignored in time-domain analysis. Moreover, the leading amplitude problem in Remark 2.3 has been demonstrated in this simulation. The leading time is almost equal to the relaxation time τ , i.e., 0.00941 ns.

In order to further demonstrate the validity, the dissipation power in the point B can be obtained by applying the integral definition for the derived and traditional power loss density, respectively. As shown in Figure 3, the non-equilibrium state has the different average dissipation powers due to irregular oscillation, which facilitates the absorb more electromagnetic energy and accelerate the rearrangement of dipole. Once the electromagnetic field in the medium approach to the equilibrium state, the dissipation powers derived by the different methods will reach an excellent agreement, which also demonstrates the explanation of Remark 2.4. Therefore, the effectiveness of transient power loss density has been successfully validated.

Figure 3. Comparison on the average dissipation power based the different transient power densities”

Point 3: What's the condition for obtaining a unique solution ?

Response 3: We apologize for the confusion. For the Maxwell's equation, the unique solution can be obtained by involving the electromagnetic boundary condition. Hence, we have added the boundary condition on the example in the last version of the manuscript, as shown:

“The left and right boundary can be defined as the perfectly matched layer, which indicates that the residual energy will be absorbed totally.”

Point 4: Please explain the affect from the central difference approximations for (3.1), (3.2) in theory and simulation.

Response 4: Thanks very much for your kind comment. In the last version of the manuscript, we have proposed the coupled Maxwell's equation and ODE model, which can be solved by applying the forward difference approximations, as shown:

“From (3.2) and (3.3), we denote that the plane wave propagates in the z direction, whose electric field orients with the x direction and magnetic field orients with the y direction. By applying the forward difference approximations for both the temporal and spatial differential operators, (3.2) and (3.3) can be transformed as

$$\varepsilon_0 \varepsilon_\infty \frac{E_x^{n+1}(k) - E_x^n(k)}{\Delta t} = - \frac{H_y^n(k) - H_y^n(k-1)}{\Delta z} - J_{px}^n(k) \quad (3.4)$$

$$\frac{H_y^{n+1}(k) - H_y^n(k)}{\Delta t} = - \frac{1}{\mu_0} \frac{E_x^n(k+1) - E_x^n(k)}{\Delta z} \quad (3.5)$$

$$\tau \frac{J_{px}^{n+1}(k) - J_{px}^n(k)}{\Delta t} + J_{px}^n(k) = \varepsilon_0 (\varepsilon_s - \varepsilon_\infty) \frac{E_{px}^{n+1}(k) - E_{px}^n(k)}{\Delta t} \quad (3.6)$$

where n and k indicate a time step $t = \Delta t \cdot n$ and a spatial distance $z = \Delta z \cdot k$.

The simulation results have been shown in Figure 2. Therefore, the last methodology is easier than the previous version of manuscript.

Reviewer 2:

Point 1: Using terms “modified Maxwell’s equations” or “modifying Maxwell’s equations” can be misunderstood. The authors are not modifying the Maxwell’s equations, instead they are inserting the constitutive relations for the Debye media, which can be considered as specifying the polarization vector in Maxwell’s equations, and they are mixing the time domain, frequency domain, and time harmonic cases.

Response 1: Thanks very much for pointing out. Based the proposed methodology, the ambiguous term has been changed as the coupled Maxwell’s equation and ordinary differential equation in the last version of the manuscript.

Point 2: For a dispersive medium, dielectric constant is not constant, therefore it would be better to call it permittivity.

Response 2: Thanks very much for your suggestion. We have changed the “dielectric constant” into “permittivity” in the last version of the manuscript.

Point 3: It should be mentioned that the equation (2.8) is given for source free region.

Response 3: Thanks very much for pointing out. We have provided the assumptions at the beginning of Section 2 in the last version of the manuscript, as shown:

“Considering the source-free, linear, isotropic and nonmagnetic Debye media, the general formulation of time-dependent Maxwell’s equations can be given as

Point 4: Equation (2.9) is given for instantaneous time [R1].

Response 4: Thanks very much for providing many authoritative literature. The derivation procedure has been given in the last version of the manuscript, as shown:

$$\begin{aligned} -\nabla \times (\mathbf{E} \times \mathbf{H}) &= \mathbf{E}(t) \frac{\partial \mathbf{D}(t)}{\partial t} + \mu_0 \mathbf{H}(t) \frac{\partial \mathbf{H}(t)}{\partial t} \\ &= \varepsilon_0 \varepsilon_\infty \mathbf{E}(t) \frac{\partial \mathbf{E}(t)}{\partial t} + \mathbf{E}(t) \frac{\partial \mathbf{D}_r(t)}{\partial t} + \mu_0 \mathbf{H}(t) \frac{\partial \mathbf{H}(t)}{\partial t} \\ &= \varepsilon_0 \varepsilon_\infty \mathbf{E}(t) \frac{\partial \mathbf{E}(t)}{\partial t} + \mu_0 \mathbf{H}(t) \frac{\partial \mathbf{H}(t)}{\partial t} \\ &\quad + \left(\frac{\tau}{\varepsilon_0 (\varepsilon_s - \varepsilon_\infty)} \frac{\partial \mathbf{D}_r(t)}{\partial t} + \frac{1}{\varepsilon_0 (\varepsilon_s - \varepsilon_\infty)} \mathbf{D}_r(t) \right) \frac{\partial \mathbf{D}_r(t)}{\partial t} \end{aligned} \quad (2.7)$$

Point 5: In figures 2 and 3, the region authors called “the unstable electromagnetic wave” is actually called transient region and time domain analysis is required (mostly) to determine this transient response.

Response 5: Thanks very much for pointing out. The statement has been modified in the last version of the manuscript, as shown:

“the transient response of electromagnetic field will be transformed as the steady response, which has the fixed amplitude and phase.”

Point 6: Please include an example that has an actual pulse excitation as in [R3].

Response 6: Thanks for your suggestion. In this paper, we focus on the analysis of transient and steady response based on the time-harmonic electromagnetic field, which can be applied in microwave heating and microwave drying. Moreover, we discover that the derived power loss density will lead the traditional one under the time-harmonic electromagnetic field, whose principle has been demonstrated by the mathematical and simulation analysis.

Point 7: The authors use $\exp(j\omega t)$ time dependence in Section 2 and in equations (3.12) and (3.13) they switched to $\exp(-i\omega t)$ time dependence without noticing.

Response 7: We apologize for the carelessness. The term “ $\exp(-i\omega t)$ ” has been deleted in the last version of the manuscript.

Point 8: It is always better to present an example that compares the proposed method with a well-known method e.g. FDTD method as in [R2].

Response 8: Thanks very much for your professional comments. In fact, the similar method in the initial manuscript can be found in [R1]. Therefore, we revise the whole manuscript and obtain a different result. Therefore, the new power loss density can be expressed as

$$P_{loss} = \frac{\tau}{\epsilon_0 (\epsilon_s - \epsilon_\infty)} \left| \frac{\partial \mathbf{D}_r(t)}{\partial t} \right|^2 \quad (2.11)$$

Hence, we have given comparison on the example in the last version of the manuscript, as shown:

“Before the simulation, we first denote the mesh size is 0.1 mm, the time step is 0.01 ps and the total time is 0.35 ns. With the help of FDTD method, the comparison results on the different transient power loss density (i.e., (2.11) and (2.12)) can be presented in Figure 2.

Figure 2 Comparison between the derived and traditional transient power loss density

When the electromagnetic field reaches the point B, the arrangement of polar molecule will be broken. The amplitudes and phases of electromagnetic wave will be changed by a large margin until the coupling process reaches equilibrium. As time goes on, the transient response of electromagnetic field will be transformed as the steady response, which has the fixed amplitude and phase. In terms of (2.11) and (2.12), the two transient power loss densities can reveal the above unsteady phenomenon. But (2.11) can reach the steady state at the relatively short time as shown in Figure 2.

Because any dynamical characteristics is not ignored in time-domain analysis. Moreover, the leading amplitude problem in Remark 2.3 has been demonstrated in this simulation. The leading time is almost equal to the relaxation time τ , i.e., 0.00941 ns.

In order to further demonstrate the validity, the dissipation power in the point B can be obtained by applying the integral definition for the derived and traditional power loss density, respectively. As shown in Figure 3, the non-equilibrium state has the different average dissipation powers due to irregular oscillation, which facilitates the absorb more electromagnetic energy and accelerate the rearrangement of dipole. Once the electromagnetic field in the medium approach to the equilibrium state, the dissipation powers derived by the different methods will reach an excellent agreement, which also demonstrates the explanation of Remark 2.4. Therefore, the effectiveness of transient power loss density has been successfully validated.

Figure 3. Comparison on the average dissipation power based the different transient power densities”

We appreciate for Editors/Reviewers’ professional works for the manuscript, and hope that the revised manuscript could satisfy the requirements for publication in this journal. Once again, thank you very much for your valuable comments and suggestions.

If you have any question about this paper, please do not hesitate to let me know.

Kind Regards,
Sincerely yours,
Jiaqi Zhong